# Identification of Four New Mutations in the GLA Gene Associated with Anderson–Fabry Disease

**DOI:** 10.3390/ijms26020473

**Published:** 2025-01-08

**Authors:** Monia Anania, Federico Pieruzzi, Irene Giacalone, Barbara Trezzi, Emanuela Maria Marsana, Letizia Roggero, Daniele Francofonte, Michele Stefanoni, Martina Vinci, Carmela Zizzo, Marcomaria Zora, Tiziana Di Chiara, Giulia Duro, Giovanni Duro, Paolo Colomba

**Affiliations:** 1Institute for Biomedical Research and Innovation (IRIB), National Research Council (CNR), 90146 Palermo, Italy; monia.anania@irib.cnr.it (M.A.); irene.giacalone@irib.cnr.it (I.G.); emanuelamaria.marsana@irib.cnr.it (E.M.M.); daniele.francofonte@irib.cnr.it (D.F.); martina.vinci@irib.cnr.it (M.V.); carmela.zizzo@irib.cnr.it (C.Z.); marcomaria.zora@irib.cnr.it (M.Z.); giovanni.duro@irib.cnr.it (G.D.); 2Nephrology Unit, Fondazione IRCCS San Gerardo dei Tintori, 20900 Monza, Italy; federico.pieruzzi@unimib.it (F.P.); barbara.trezzi@unimib.it (B.T.); letizia.roggero@unimib.it (L.R.); 3School of Medicine and Surgery, University of Milano-Bicocca, 20126 Milan, Italy; michele.stefanoni@unimib.it; 4Excellence Department of Health Promotion, Maternal and Child, Internal and Specialist Medicine “G. D’Alessandro”, University of Palermo, 90127 Palermo, Italy; tiziana.dichiara@unipa.it; 5Internal Medicine, Ospedale Cattinara, 34149 Trieste, Italy; giulia.duro@libero.it

**Keywords:** Fabry disease, *GLA* gene, α-galactosidase A, Lyso-Gb3, novel mutations

## Abstract

Anderson–Fabry disease is a hereditary, progressive, multisystemic lysosomal storage disorder caused by a functional deficiency of the enzyme α-galactosidase A (α-GalA). This defect is due to mutations in the *GLA* gene, located in the long arm of the X chromosome (Xq21-22). Functional deficiency of the α-GalA enzyme leads to reduced degradation and accumulation of its substrates, predominantly globotriaosylceramide (Gb3), which accumulate in the lysosomes of numerous cell types, giving rise to the symptomatology. Clinical diagnosis can still be difficult today due to the peculiarities of the disease, which presents with clinical manifestations that overlap with those of other pathologies and a wide possibility of differential diagnoses, which lead to missed diagnoses, misdiagnosis, or a diagnostic delay. Patients with clinical suspicion of Fabry disease undergo a diagnostic workup that includes an evaluation of α-GALA enzyme activity, genetic analysis of the *GLA* gene, and the measurement of blood Lyso-Gb3, a soluble derivative of Gb3. In this paper, we describe four novel mutations identified in the *GLA* gene which are associated with absent or reduced α-GalA activity, pathological accumulation of the specific substrate, and characteristic clinical manifestations of Fabry disease. We identified two mutations (c.583insGAATA and p.Y207X) that result in the formation of a premature translation stop codon, resulting in a truncated protein and thus a completely non-functional enzyme. The other two identified gene alterations (p.G261C and c.786G>T, which determine p.W262C) are missense mutations that cause reduced α-GALA activity, the accumulation of blood Lyso-Gb3, and symptoms consistent with Fabry disease, and therefore may be associated with this disorder. The identification of these new mutations in patients with symptoms attributable to Fabry disease increases the molecular knowledge of the *GLA* gene and provides important support to the clinician, for a more accurate and timely diagnosis of the pathology.

## 1. Introduction

Anderson–Fabry disease (or Fabry disease, OMIM #301500) is a progressive, hereditary, multisystemic lysosomal storage disorder characterized by functional deficiency of the enzyme α-galactosidase A (α-GAL A) [1]. This deficit determines an alteration in the metabolism of some glycosphingolipids, predominantly globotriaosylceramide (Gb3), which accumulates in the lysosomes of numerous cell types, especially in vascular endothelial cells [2]. The degradation of other substrates is also impaired, but their contribution to disease is not well known [3].

From an etiological point of view, Fabry is an X-linked lysosomal enzymopathy determined by mutations in the *GLA* gene, which encodes α-GALA [2,4], located on the long arm of the X chromosome (Xq21-22).

The suspicion is advanced on the basis of clinical data, as well as anamnestic and family history, and then confirmed through genetic and biochemical analyses, such as the identification of the gene alteration and the measurement of α-galactosidase A activity, which may be null or strongly reduced [5]. The determination of the enzyme substrates, Gb3 inside cells and its deacylated derivative, Lyso-Gb3 (globotriaosylsphingosine), at blood level, also constitutes an important form of diagnostic support. The accumulation of Gb3, with the formation of characteristic lamelliform inclusions at the level of lysosomes, can be found in various cell types, including vascular cells, endothelial cells, cardiomyocytes, podocytes, and neurons in ganglia and in the central nervous system [4]. The assessment of Gb3 levels by biopsy and the measurement of circulating Lyso-Gb3 represent a reliable methodology for patient monitoring and treatment efficacy. The dosage of Lyso-Gb3 in plasma is used to “certainly” identify patients affected by FD, proving to be a reliable diagnostic marker [6]. A reduction in GB3 and Lyso-Gb3 is often observed in patients undergoing enzyme replacement therapy (ERT), and is also useful in patient follow-up and to evaluate the efficacy of the therapy [7]. Currently available treatment options for Fabry disease include enzyme replacement therapy (ERT) and chaperone therapy, both supported by other therapies to manage symptoms.

The first symptoms can appear at different ages, generally in childhood, and have different severity and progression [8]. The classic clinical picture of Fabry usually appears in childhood or adolescence with angiokeratomas (especially in the “swimsuit” area), corneal opacity, microalbuminuria or proteinuria, and some impairment of the autonomic nervous system. As age advances, the disease leads to significant impairment of the renal and cardiovascular systems. The accumulation of glycosphingolipids in renal cells compromises their function, leading to progressive nephropathy, which is one of the most critical features of Fabry [4].

In males with the classic form of Fabry disease, α-GalA activity is very low or absent: diagnosis is reliably made by measuring the enzyme’s activity in dried blood spots (DBFP assay), plasma, or isolated leukocytes [4]. Female individuals, usually heterozygous, whose organs are chimeras of normal and affected cells, due to the lyonization of the X chromosome, generally have less evident symptoms that are more difficult to detect [9]. However, females can also present cardio and cerebrovascular disorders of the same severity as hemizygous males, following the imbalance of X chromosome inactivation [10]. Furthermore, always due to lyonization, in females, the activity of α-GalA is extremely variable, and therefore less reliable as a first diagnostic step, oscillating between normal and pathological values, even in the presence of causative mutations; the differences observed in the enzymatic activity between males and females have also been found in the accumulation of Lyso-Gb3 in the blood. Thus, the blood Lyso-Gb3 assay in females is similarly unreliable. Consequently, in these subjects, genetic analysis turns out to be the only reliable and indispensable diagnostic means for a correct diagnosis of Fabry disease [11].

In Anderson–Fabry disease, in addition to the classic phenotype, an increasing number of individuals with an attenuated phenotype, the so-called atypical variants, have been identified [12,13]. These usually manifest as a mild, late-onset phenotype and nonspecific symptoms that make diagnosis particularly difficult. For this reason, in subjects with these mutations the disease is diagnosed in adulthood, when the organ damage that has occurred is irreversible [14]. Diagnosing this pathology can be difficult due to its peculiarities. Furthermore, the clinical manifestations overlap with those of other pathologies with a high possibility of differential diagnoses, which involve different medical specializations. Diagnostic error is a concrete risk that determines an underestimation of the real number of affected subjects [15].

To date, more than 1000 mutations in the *GLA* gene have been described in Fabry patients [16]. However, the genotype–phenotype correlation is not always clear in this disease [17], also because of the wide variability in clinical manifestations. Phenotypic heterogeneity is mainly expressed in two forms of the disease, classic and late-onset, although today, many variants still remain undescribed in the literature [18]. There is little genotype–phenotype correlation even at the intrafamilial level, and this contributes to a wide variety of symptoms, often overlapping with other diseases as well [19]. Future studies may help to understand the reasons for this variability; one of the factors is probably epigenetics. In Fabry disease, the identification of new mutations is therefore important to associate them with new phenotypes [19,20].

In this work, we describe four new mutations identified in the *GLA* gene of patients with a clinical diagnosis attributable to FD. The identification of new mutations increases the molecular knowledge of the *GLA* gene and provides the clinician with support for a correct diagnosis of the disease. The results reported here suggest that these new mutations may be related to the classic form of Fabry disease. Furthermore, genetic investigation of relatives of affected patients also allows us to identify pre-symptomatic subjects who can start therapy as early as possible, before organ damage can become irreversible.

## 2. Results

In this work we describe four novel mutations (p.G261C and c.786G>T, which determine p.W262C; c.583insGAATA; and p. Tyr207X) that we identified in patients with clinical suspicion of Fabry disease. In the presence of symptoms attributable to the disease, we proceeded with a complete diagnostic process that included the following: an evaluation of the enzymatic activity of α-galactosidase A, genetic analysis of the *GLA* gene, and the determination of blood Lyso-Gb3. The genetic alterations we found in these patients are not present in the Human Gene Mutation Database (http://www.hgmd.org, accessed on 9 December 2024) and in the Fabry disease databases. We named the four mutations according to the mutation naming guidelines recommended by the Human Genome Variation Society (www.HGVS.org/varnomen, accessed on 9 December 2024).

### 2.1. CASE 1: Mutation p.G261C

The proband was a 51-year-old woman with modest proteinuria and albuminuria and suspected cardiovascular damage; the woman showed conduction defects, tortuosity of the retinal blood vessels, hypertension, and a TIA (transient ischemic attack) event. She also had cornea verticillata, typical of Fabry disease.

We then proceeded with the analysis of the seven exons of the *GLA* gene and the intronic regions flanking the exons. In exon 5 of the gene, we identified the novel missense mutation c.781G>T (Figure 1a), which causes the substitution of a Glycine with a Cysteine at position 261 of the protein (p.G261C). This mutation has not been previously described in the literature and is not reported in Fabry disease mutation databases, but a different mutation that falls at the same site, G261D, has been found in patients with classic FD.

The patient’s enzyme activity was 1.4 nmol/mL/h, below the normal values for α-galactosidase A activity (normal reference values above 3 nmol/mL/h), and a Lyso-Gb3 value of 2.31 nmol/L (normal reference values below 2.3 nmol/L) (Table 1).

The study was also extended to the patient’s 54-year-old brother and 53-year-old sister (Table 1). The brother presented a typical clinical picture of FD: acroparesthesias in childhood, abdominal pain with gastrointestinal manifestations, intolerance to heat and/or cold, left ventricular hypertrophy (LVH), tortuosity of the retinal blood vessels, hearing loss, and hypertension. The sister had LVH, hypertension, hearing loss, and cornea verticillata. Fabry disease had not been previously considered for either. In the samples of these two subjects, we identified the same novel mutation (in hemizygosis in the brother, Figure 1b), with almost null enzymatic activity in the male patient (0.7 nmol/mL/h), and of 1.8 nmol/mL/h in the female patient, with blood Lyso-Gb3 values of 29.9 nmol/L and 4.88 nmol/L, respectively (Table 1).

### 2.2. CASE 2: Mutation c.786G>T, Which Determines p.W262C

The proband was a 41-year-old male with events of infantile acroparesthesias, hypo-anhidrosis, proteinuria angiokeratomas, and recurrent fever. The clinical suspicion of Fabry disease suggested proceeding with the determination of enzymatic activity, which was found to be almost null (0.1 nmol/mL/h, normal reference values greater than 3 nmol/mL/h) (Table 2). Genetic analysis of the *GLA* gene revealed a hemizygous Guanine–Thymine variation in exon 5, at position 786 of the cDNA (c.786G>T), which determines the substitution of a Tryptophan with a Cysteine at position 262 of the protein (p.W262C) (Figure 2a). The nucleotide variation c.786G>T has not been previously described in the literature and is not reported in Fabry disease-associated mutation databases, but determines the same amino acid substitution W262C that is instead reported as classic, and is associated with the mutation c.786G>A [21]. In the blood, Lyso-Gb3 was found to be very high (93.27 nmol/L, normal reference values less than 2.3 nmol/L) (Table 2).

The molecular study extended to the patient’s mother, aged 67 years old (Table 2), and revealed the same mutation in heterozygosis (Figure 2b), with normal enzymatic activity (plausible for a female, due to lyonization), but with a high blood Lyso-Gb3 value (12.34 nmol/L).

### 2.3. CASE 3: Mutation c.583insGAATA

Patient 3 was a 49-year-old female with a very complex clinical situation. She was undergoing dialysis replacement therapy, with a compromised renal picture: albuminuria, proteinuria, hyperazotemia. She also presented with cardiac/circulatory aspects such as conduction defects, left ventricular hypertrophy, and recurrent headaches, in addition to events of acroparesthesias and angiokeratomas, typical of FD.

Genetic analysis of the *GLA* gene allowed us to identify a five-nucleotide insertion (c.583insGAATA) in exon 4 of the gene (Figure 3), which causes slippage of the reading frame of translation, with the substitution of a Glycine with a Glutamic Acid at position 195 of the protein and the formation of a premature translation stop codon one nucleotide downstream from the mutation site (p.Gly195Glufs*1). This results in the synthesis of a truncated protein of 195 amino acids, instead of 429, and consequently, a non-functional enzyme.

The DBFP test showed normal enzyme activity of 4.6 nmol/mL/h (normal reference values greater than 3 nmol/mL/h), plausible for a female due to lyonization, and a pathological blood Lyso-Gb3 value (12.74 nmol/L, normal reference values less than 2.3 nmol/L) (Table 3).

The mutation is not reported in Fabry disease mutation databases and has not been previously described in the literature, but the presence of blood accumulation of Lyso-Gb3 found in this patient, as well as the nature of this mutation itself, suggest its pathogenic role.

### 2.4. CASE 4: Mutation p.Y207X+p.A143T

Patient 4 was a 39-year-old female who presented with angiokeratomas, intolerance to heat or cold, and the typical cornea verticillata of FD.

Genetic analysis of the *GLA* gene highlighted the novel nonsense mutation in exon 4 of the gene (c.621T>A) (Figure 4a): a change in a Thymine to an Adenine at position 621 of the *GLA* gene cDNA, which introduces a stop codon at position 207 of the polypeptide chain, generating a truncated protein (p.Y207X). In addition, the mutation, known as GVUS, c.427G>A, p. A143T [22,23,24], a variant that has a conflicting pathogenicity classification, was also found (Figure 4b).

The enzyme activity analysis result was 3.2 nmol/mL/h, just above the cut-off value (normal reference values greater than 3 nmol/mL/h), plausible for a female, due to lyonization (Table 4). The blood Lyso-Gb3 value was pathological, with 9.92 nmol/L (normal reference values less than 2.3 nmol/L) (Table 4).

The study was also extended to the patient’s mother, aged 74, in whom the same mutations were found, which are evidently located on the same X chromosome in cis. Her enzymatic activity was found to be normal (18.6 nmol/mL/h), while her blood Lyso-Gb3 was found to be pathological, with a value of 7.89 nmol/L (Table 4). The patient’s clinical study is currently being investigated.

## 3. Discussion

The early diagnosis of patients with Fabry disease is extremely important to take action promptly with the most appropriate therapy [25]. Mutation analysis in the *GLA* gene is a fundamental step in the diagnosis of this disease in patients with clinical suspicion [26]. To date, more than 1000 mutations in the *GLA* gene responsible for Fabry disease have been described, including missense, nonsense, and small and large deletions/insertions. In this work, we described four novel mutations (p.G261C and c.786 G>T, which determine p.W262C; c.583insGAATA; and p.Tyr207X) that we identified in unrelated patients, confirming the heterogeneity observed in subjects affected by FD. The discovery of new genetic variants is essential to expand the database of known mutations and to improve genetic diagnosis. Where possible, we have extended genetic and enzymatic analysis to the patients’ relatives, to make a significant contribution to the understanding of the variants. The extension of genetic and enzymatic analysis to relatives is particularly important to identify carriers or asymptomatic affected subjects, allowing preventive treatment and more careful clinical monitoring. The insertion (c.583insGAATA), the nonsense mutation (p.Tyr207X), and the missense mutations (p.G261C and c.786 G>T, which determine p.W262C) described in this article can be considered responsible for Fabry disease, due to the nature of the mutation and/or the pathological values of α-galactosidase A enzyme activity and/or blood Lyso-Gb3 found in patients.

The case 1 patients have a Glycine–Cysteine substitution in codon 261 of the protein (p.G261C). This can alter the structure of the protein, compromising its function and thus causing Fabry disease: all subjects in the family study have enzymatic activity below the reference range, resulting in substrate accumulation and the distinctive clinical signs of the disease. According to a study using Polyphen 2 (http://genetics.bwh.harvard.edu/pph2/, accessed on 9 December 2024), a tool that allows us to predict the possible effect of an amino acid substitution on the structure and function of a human protein, we can predict that the mutation could be harmful, with a score of 1 (in the range from 0, benign, to 1, pathological). Furthermore, it is shown that the amino acid G261 is highly conserved in evolution (Figure 5). The combination of clinical, genetic, biochemical, and predictive data supports the hypothesis that the p.G261C mutation could play a key role in Fabry disease, as well as another mutation, p.G261D, which falls in the same position of the protein and has been previously described in patients affected by the classic form of Fabry disease [27].

The patients of case 2 present the substitution of a Tryptophan, in position 262 of the protein, with a Cysteine (p.W262C). The proband, a male, presents almost null enzymatic activity, with notable values of blood Lyso-Gb3; the mother presents normal enzymatic activity (in accordance with the principle of Lyonization) and elevated blood Lyso-Gb3, in addition to the presence of the classic clinical signs of the disease. Again, the study using Polyphen 2 reveals that the mutation could be harmful, with a score of 1, also due to the conservation of the amino acid Tryptophan at position 262 in evolution (Figure 6). The combination of clinical, genetic, biochemical, and predictive data supports the hypothesis that the nucleotide substitution c.786G>T could play a key role in Fabry disease, as another mutation, c.786G>C, which falls in the same position of the cDNA, causes the same amino acid substitution W262C and has been previously described in patients affected by the classic form of Fabry disease [22,28]. In the Fabry_CEP database (https://web.na.icb.cnr.it/fabry_cep/, 9 December 2024), this variant has been tested; the replacement does not occur at the active site and does not disrupt disulfide bonds and is not amenable to pharmacological chaperones.

The mutations in cases 3 and 4 (c.583insGAATA and c.621T>A) are responsible for the introduction of a premature stop codon, resulting in the formation of a truncated and non-functional protein, and have been found in patients affected by the classic FD phenotype.

The mutation of case 3, found in exon 4 of the GLA gene, is a 5 bp insertion (GAATA) at position 583 of the cDNA, which causes slippage of the reading frame, the substitution of a Glycine with a Glutamic Acid at position 195 of the protein, and the formation of a translation stop codon one triplet later. This determines the synthesis of a truncated protein of 195 amino acids, instead of 429, and consequently renders the protein non-functional. The presence of blood accumulation of Lyso-Gb3 found, and the nature of the mutation itself, in association with the clinical signs, suggest its pathogenetic role.

The patients in case 4 present the replacement of a Thymine with an Adenine in position 621 of the cDNA, which determines the formation of the UAA stop codon in position 207, giving rise to the premature termination of translation, and therefore to a protein of 206 amino acids, instead of 429. Also, in this case, the presence of high blood Lyso-Gb3 and the nature of the mutation itself would suggest that this mutation leads to a significant loss of protein function, and therefore to the development of the disease. In these patients, the p.A143T mutation was also found in the *GLA* gene, the involvement of which in the symptoms is still debated today [22,23,29]. This mutation is usually found in patients who, despite presenting some symptomatic aspects attributable to Fabry disease, have generally normal α-galactosidase A activity, and do not have pathological values of blood Lyso-Gb3. Therefore, in the two patients described here, the elevated values of Lyso-Gb3 and the symptoms are probably attributable to the p.Y207X mutation, so p.A143T does not affect the phenotype of patients.

The fact that in most of the female patients described in this article, the activity of the enzyme α-galactosidase A is normal, should not be surprising. This, in fact, is normal and is due to the localization of the GLA gene on the X chromosome, and to the phenomenon of lyonization, the inactivation of one of the two X chromosomes in the different cells of the female organism. It has been observed that in some females, the inactivation of the X chromosome is not random in all cells, but some of them show a phenomenon of asymmetric inactivation [30]. This causes great clinical heterogeneity, ranging from the absence of manifestations to severity of the clinical picture comparable to that of males affected by Fabry disease [29,31,32]. This heterogeneity can also be observed in the values of enzymatic activity, which is normal in more than 70% of female patients, and of blood Lyso-Gb3, which is normal in 80% of female patients with a late-onset phenotype and in 17% of females with a classic phenotype [11]. The activity values of α-Gal A and Lyso-Gb3 found in the patients described in this article reflect these percentages.

All patients studied in this article show symptoms associated with the classic form of Fabry disease. However, all the cases presented were diagnosed in adulthood when one or more appreciable clinical manifestations, such as stroke, renal failure, or left ventricular hypertrophy, led clinicians to hypothesize that the patients had Fabry disease.

## 4. Materials and Methods

### 4.1. Patients

Peripheral blood was collected, using EDTA as an anticoagulant, and dried on absorbent paper (dried blood spot, DBS). Genetic and enzymatic studies were performed at the Centre for Research and Diagnosis of Lysosomal Storage Disorders of IRIB-CNR in Palermo.

### 4.2. Genetic Analysis

Genomic DNA was isolated from the dried blood spot using silica-coated magnetic particles, in a robotic workstation for the automated purification of nucleic acids (Qiagen, Hilden, Germany). DNA concentrations were estimated using a biophotometer (Eppendorf, Hamburg, Germany). The search for mutations in the GLA gene was performed through Sanger sequencing. Eight pairs of primers were designed to analyze eight target regions containing the seven exons of the *GLA* gene, including the flanking regulatory sequences, and the cryptic exon. PCR products were purified and sequenced at Eurofins Genomics (Ebersberg, Germany).

### 4.3. α-Galactosidase A Activity Assay

α-galactosidase A activity assays were performed using the dried-blood filter paper (DBFP) test described by Chamoles et al. [33], with modifications [unpublished data]. A spot of 10 μL of blood on a circle of paper 6 mm in diameter was placed into a 96-well plate, suitable for fluorometric assays, and incubated for 18 h at 37 °C in a thermomixer; the reaction was terminated by the addition of 250 μL of 0.1 mol/L ethylenediamine (pH 11.4). The background fluorescence, i.e., fluorescence which was not due to specific enzyme activity, was determined for each sample, conducting another reaction in the presence of 0.14 mmol/l 1-deoxygalactonojirimycin (DGJ, the inhibitor of alpha galactosidase A) in citrate phosphate buffer (pH 4.5). This background was subtracted from the fluorescence of the sample. In each assay, we added positive and negative controls and a calibration curve with 4-methylumbelliferone. Normal values were considered to be > 3.0 nmol/mL/h.

### 4.4. Lyso-Gb3 Determination

The determination of Lyso-Gb3 in blood was performed using the tandem mass spectrometry (MS/MS) methodology as previously described by Polo et al. [34].

## 5. Conclusions

To date, Fabry disease is still difficult to diagnose, as its symptoms often overlap with those of other diseases. Diagnostic error is therefore a real risk, which could lead to a delay in diagnosis, and thus a delay in the start of therapeutic treatment. From a clinical point of view, Fabry disease should be considered in all patients with symptoms attributable to the disease, uncertain diagnosis, and an atypical clinical course or a clinical picture with an unclear systemic implication. This clinical approach would help to limit misdiagnosis between Fabry disease and other diseases or cases of missed diagnosis. This could also help to reduce the time between the onset of symptoms and the diagnosis of FD, avoiding treatments that are not useful for patients and starting the available therapy specific to the disease they are affected by.

The results described here suggest that these four new mutations may be correlated with the classic manifestations of Fabry disease, as the described cases show reduced enzyme activity (in male subjects), the accumulation of Lyso-Gb3, and typical signs of the disease. These new data may help clinicians in the diagnosis of the disease, and increase the clinical and molecular knowledge on the correlation between GLA gene mutations and the disease phenotype.

## Figures and Tables

**Figure 1 ijms-26-00473-f001:**
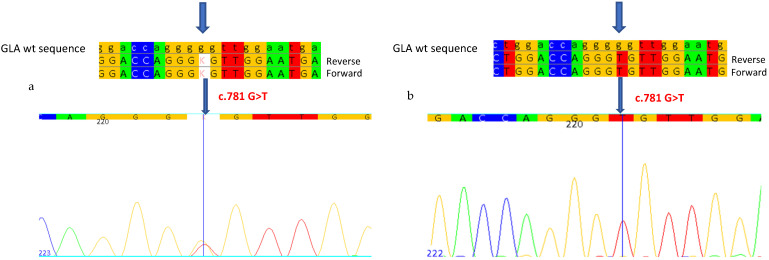
Sanger sequencing shows the c.781G>T mutation in a heterozygous state in the proband (**a**) and in a hemizygous state in the patient’s brother (**b**). Each base corresponds to a color: red for Thymine, blue for Cytosine, yellow for Guanine, green for Adenine. K indicates the mutation in the heterozygous state as Thymine. Arrows indicate the mutation.

**Figure 2 ijms-26-00473-f002:**
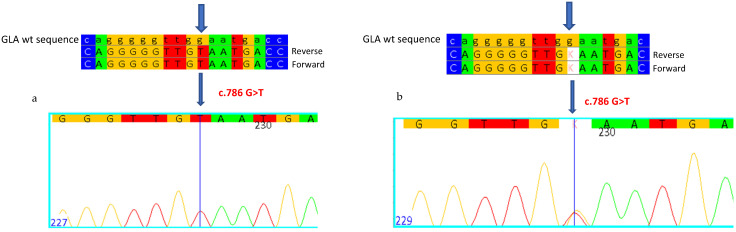
Sanger sequencing shows the c.786G>T mutation in hemizygosity in the proband (**a**) and in heterozygosity in his mother (**b**). Each base corresponds to a color: red for Thymine, blue for Cytosine, yellow for Guanine, green for Adenine. K indicates that the mutation in heterozygosity is Thymine. Arrows indicate the mutation.

**Figure 3 ijms-26-00473-f003:**
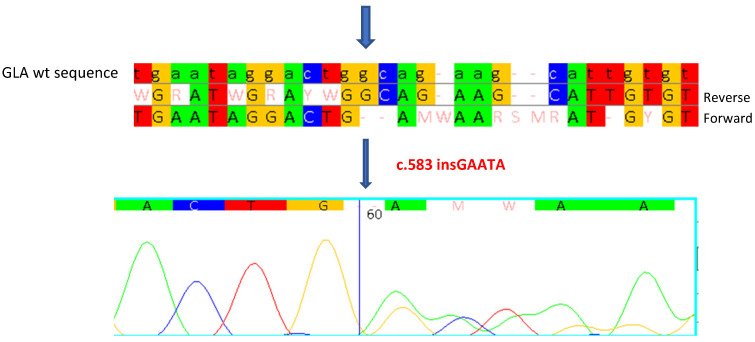
Sanger sequencing shows the c.583insGAATA mutation in the heterozygous state in the proband. Each base corresponds to a color: red for Thymine, blue for Cytosine, yellow for Guanine, green for Adenine. Arrows indicate the insertion start point.

**Figure 4 ijms-26-00473-f004:**
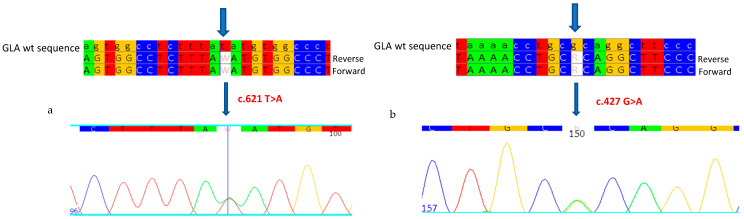
Sanger sequencing shows the mutations c.621T>A (**a**) and c.427G>A (**b**) in the heterozygous state in the proband. Each base corresponds to a color: red for Thymine, blue for Cytosine, yellow for Guanine, green for Adenine. W and R indicate that the mutations in heterozygosity are Adenine. Arrows indicate the mutations.

**Figure 5 ijms-26-00473-f005:**
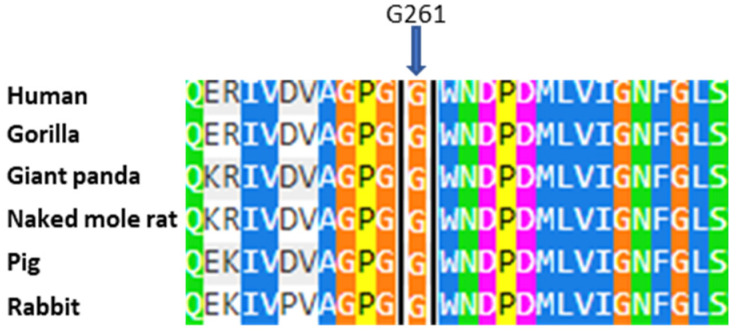
Alignment of human α-Gal A amino acids with sequences from other organisms. The arrow indicates the amino acid at position 261 that was found to be mutated in the case described. Each color corresponds to a different percentage of amino acid conservation in different species.

**Figure 6 ijms-26-00473-f006:**
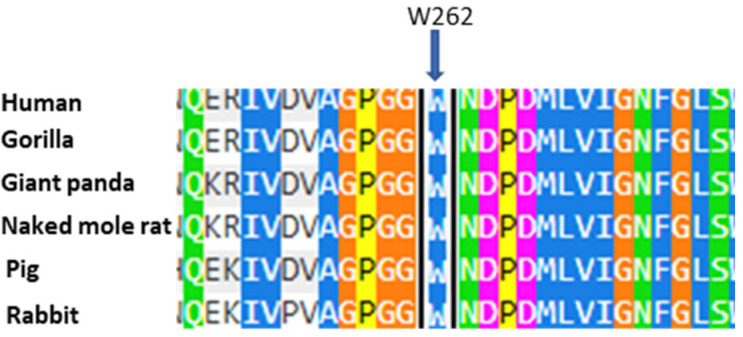
The alignment of the amino acids of human α-GalA with sequences from other organisms. The arrow indicates the amino acid at position 262 that was found to be mutated in the case described. Each color corresponds to a different percentage of amino acid conservation in different species.

**Table 1 ijms-26-00473-t001:** Molecular and clinical data of patients with the p.G261C mutation (TIA: transient ischemic attack; LVH: left ventricular hypertrophy). Bold indicates pathological values.

Patient	Sex	Age	Mutation	Alpha-Galactosidase A Activity (nmol/mL/h) Normal Range: >3.0 nmol/mL/h	Lyso-Gb3 in Blood (nmol/L) Normal Range: <2.3 nmol/L	Clinical Signs
Proband	F	51	p.G261C	**1.4**	**2.31**	Conduction defects, tortuosity of retinal blood vessels, hypertension, TIA event, cornea verticillata, albuminuria, and proteinuria
Brother	M	54	p.G261C	**0.7**	**29.9**	Acroparesthesia in childhood, abdominal pain with gastrointestinal manifestations, intolerance to heat and/or cold, LVH, tortuosity of retinal blood vessels, hearing loss, and hypertension
Sister	F	53	p.G261C	**1.8**	**4.88**	LVH, hypertension, hearing loss, and cornea verticillata

**Table 2 ijms-26-00473-t002:** Molecular data of patients with the p.W262C mutation (LVH: left ventricular hypertrophy). Bold indicates pathological values.

Patient	Sex	Age	Mutation	Alpha-Galactosidase A Activity (nmol/mL/h) Normal Range: >3.0 nmol/mL/h	Lyso-Gb3 in Blood (nmol/L) Normal Range: <2.3 nmol/L	Clinical Signs
Proband	M	41	p.W262C	**0.1**	**93.27**	Infantile acroparesthesia, hypo-anhidrosis, angiokeratoma, recurrent fever
Mother	F	67	p.W262C	5.8	**12.34**	Hypo-anhidrosis, LVH

**Table 3 ijms-26-00473-t003:** The molecular data of the patient with the c.583insGAATA mutation. (LVH: left ventricular hypertrophy). Bold indicates pathological values.

Patient	Sex	Age	Mutation	Alpha-Galactosidase A Activity (nmol/mL/h) Normal Range: >3.0 nmol/mL/h	Lyso-Gb3 in Blood (nmol/L) Normal Range: <2.3 nmol/L	Clinical Signs
Proband	F	49	c.583insGAATA	4.6	**12.47**	Renal replacement treatment, albuminuria, proteinuria, azotemia, conduction defects, LVH, recurrent headaches, acroparesthesias, angiokeratomas

**Table 4 ijms-26-00473-t004:** Molecular data of patients with the double mutation A143T+Y207X. Bold indicates pathological values.

Patient	Sex	Age	Mutation	Alpha-Galactosidase A Activity (nmol/mL/h) Normal Range: >3.0 nmol/mL/h	Lyso-Gb3 in Blood (nmol/L) Normal Range: <2.3 nmol/L	Clinical Signs
Proband	F	39	p.A143T + p.Y207X	3.2	**9.92**	Angiokeratoma, heat or cold intolerance, cornea verticillata
Mother	F	74	p.A143T + p.Y207X	18.6	**7.89**	Clinical study in progress

## Data Availability

The data are contained within the article.

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
