# Peer review of "Identification of Four New Mutations in the GLA Gene Associated with Anderson–Fabry Disease"

_ijms, 2025, doi:10.3390/ijms26020473_

Round 1
Reviewer 1 Report
Comments and Suggestions for Authors
The paper “
Identification of four new mutations in the GLA gene referable to Anderson-Fabry disease “
Is meritorious because it presents interesting cases that help demonstrate the complexity of diagnosing and treating Fabry disease. However, it requires major revisions before being acceptable for publication.
Major:
First point:
The authors claim:
“In this work we describe four novel mutations (p.G261C, p.W262C, c.583insGAATA 155 and p. Tyr207X) that we identified in patients with clinical suspicion of Fabry disease. In the presence of symptoms attributable to the disease, we proceeded with a complete diagnostic process that includes: evaluation of the enzymatic activity of α-galactosidase A, genetic analysis of the GLA gene and determination of blood Lyso-Gb3. The genetic alterations we found in these patients are not present in the Human Gene Mutation Database (http://www.hgmd.org), and in the Fabry disease databases. We named the four mutations according to the mutation naming guidelines recommended by the Human Genome Variation Society. (www.hgv.org/mutnomen). “
This misleading. In fact, as the same authors precise later, it is the nucleotide variant to be new, but not the protein variant:
“Genetic analysis of the GLA gene revealed a hemizygous Guanine Thymine variation in exon 5, at position 786 of the cDNA (c.786G>T), which determines the substitution of a Tryptophan with a Cysteine at position 262 of the protein (p.W262C) (Figure 2a). The nucleotide variation c.786G>T has not been previously described in the literature and is not reported in Fabry disease-associated mutation databases, but determines the same amino acid substitution W262C that is instead reported as classic, and associated with the mutation c.786G>A (23).”
Moreover, they claim the variants are absent in the Fabry disease databases. On the contrary, p.W262C is reported in Fabry_CEP, where it is specified that the variant has been tested, does not occur in the active site, does not disrupt disulphide binds, and is not amenable to therapy with pharmacological chaperones.
The database should be cited (https://doi.org/10.1186/1750-1172-8-111 ) and amenability to pharmacological chaperones, too.
Second point:
p.Ala143Thr, a relatively frequent variant according to GnomAD (5.81e-4), has a conflicting pathogenicity classification. However, since it is cis with a non-sense mutation, it cannot influence patient 4’s phenotype. A truncated protein simply does not work or does not exist in the cell, and a concomitant missense mutation has no effect. This should be clarified.
Minor:
GLA gene in italics
Consistency lysoGb3 lyso-Gb3?
Reviewer 2 Report
Comments and Suggestions for Authors
The manuscript provides valuable insights by identifying four novel GLA mutations linked to Fabry disease. The findings enhance molecular understanding and aid in clinical diagnosis. This manuscript is well-prepared and offers significant contributions to the field. With these revisions, it will be a valuable addition to the literature.
1. In material and methods, authors have mentioned Galactosidase A Activity Assay, in which its written in line number 141, with modifications [unpublished data]. Can authors elaborate this part?
2. Human Genome Variation Society. (www.hgv.org/mutnomen). Provided link is not working. It must be corrected.
3. In figure1, figures number a and b are not clear. It has to be corrected. Same goes to another figures. However, consider increasing the resolution of these figures for improved clarity.
4. The description of how the mutations (c.583insGAATA and p.Y207X) lead to the formation of a premature stop codon and a truncated protein is clear. But does not provide evidence or references to support the assertion that these mutations lead to a completely non-functional enzyme. It should be addressed.
Reviewer 3 Report
Comments and Suggestions for Authors|
Title: Identification of four new mutations in the GLA gene referable to Anderson-Fabry disease Authors: Monia Anania, Federico Pieruzzi, Irene Giacalone, Barbara Trezzi, Emanuela Maria Marsana, Letizia Roggero, Daniele Francofonte, Michele Stefanoni, Martina Vinci, Carmela Zizzo, MarcoMaria Zora, Tiziana Di Chiara, Giulia Duro, Giovanni Duro and Paolo Colomba |
|
|
|
|
The paper aimed to these four new mutations might be correlated with the classic manifestations of Fabry disease. These new data may help the clinician in the diagnosis of the disease, and increase the clinical and molecular knowledge in the correlation between GLA gene mutations and the disease phenotype.
(1) “2.3α-. Galactosidase A Activity Assay”
The “.”after “α-”should be deleted.
(2) The content studied in the paper is too simplistic.
Please add some evaluation of α-GAL A enzyme activity, genetic analysis of the GLA gene, and measurement of blood Lyso-Gb3, a soluble derivative of Gb3 and others.
Round 2
Reviewer 1 Report
Comments and Suggestions for Authors
The authors have addressed the major points requested, but for correctness, they should cite Fabry_CEp in the references as they do for other databases. The correct way to cite can be found in google scholar using the doi and is
Cammisa, Marco, et al. "Fabry_CEP: a tool to identify Fabry mutations responsive to pharmacological chaperones." Orphanet Journal of Rare Diseases 8 (2013): 1-3
Reviewer 3 Report
Comments and Suggestions for Authors
The manuscript titled "Identification of four new mutations in the GLA gene referable 2 to Anderson-Fabry disease" can be acceepted.